# Partial label learning for automated classification of single-cell transcriptomic profiles

**Malek Senoussi**[1], **Thierry Artieres**[1,2]*, **Paul Villoutreix**[1,3]*

**1** Aix Marseille Univ, Université de Toulon, CNRS, LIS, Turing Centre for Living Systems, Marseille, France, **2** Ecole Centrale de Marseille, Marseille, France, **3** Aix-Marseille Université, MMG, Inserm U1251, Turing Centre for Living systems, Marseille, France

* thierry.artieres@univ-amu.fr (TA); paul.villoutreix@univ-amu.fr (PV)

## Abstract

Single-cell RNA sequencing (scRNASeq) data plays a major role in advancing our understanding of developmental biology. An important current question is how to classify transcriptomic profiles obtained from scRNASeq experiments into the various cell types and identify the lineage relationship for individual cells. Because of the fast accumulation of datasets and the high dimensionality of the data, it has become challenging to explore and annotate single-cell transcriptomic profiles by hand. To overcome this challenge, automated classification methods are needed. Classical approaches rely on supervised training datasets. However, due to the difficulty of obtaining data annotated at single-cell resolution, we propose instead to take advantage of partial annotations. The partial label learning framework assumes that we can obtain a set of candidate labels containing the correct one for each data point, a simpler setting than requiring a fully supervised training dataset. We study and extend when needed state-of-the-art multi-class classification methods, such as SVM, kNN, prototype-based, logistic regression and ensemble methods, to the partial label learning framework. Moreover, we study the effect of incorporating the structure of the label set into the methods. We focus particularly on the hierarchical structure of the labels, as commonly observed in developmental processes. We show, on simulated and real datasets, that these extensions enable to learn from partially labeled data, and perform predictions with high accuracy, particularly with a nonlinear prototype-based method. We demonstrate that the performances of our methods trained with partially annotated data reach the same performance as fully supervised data. Finally, we study the level of uncertainty present in the partially annotated data, and derive some prescriptive results on the effect of this uncertainty on the accuracy of the partial label learning methods. Overall our findings show how hierarchical and non-hierarchical partial label learning strategies can help solve the problem of automated classification of single-cell transcriptomic profiles, interestingly these methods rely on a much less stringent type of annotated datasets compared to fully supervised learning methods.

**Data Availability Statement:** Code and data are available on the following GitHub repository: https://github.com/paulvill/plRNAseq.

**Funding:** MS and PV were employees of Aix-Marseille University and funded by the

"Investissements d'Avenir" French Government program managed by the French National Research Agency (ANR-16-CONV-0001) and from Excellence Initiative of Aix-Marseille University - A*MIDEX. TA was an employee of Ecole Centrale de Marseille. This work was performed using HPC resources from GENCI-IDRIS (Grant 20XX-AD011013899). The funders had no role in study design, data collection and analysis, decision to publish, or preparation of the manuscript.

**Competing interests:** The authors have declared that no competing interests exist.

## Author summary

Recent years have witnessed an exponential increase in the amount of single-cell RNASeq data generated, particularly in studies of development. One of the major challenges is to identify individual cell types within the data. Expert knowledge is required to identify the relevant marker genes, tissue and timing that will enable the cell type identification. This information can be difficult to obtain and calls for automated cell type classification approaches. Classical classification techniques would solve this problem by training a classifier on a fully supervised dataset. However, this only pushes the problem further, as a dataset annotated at single-cell resolution is still needed for training. Here we propose instead to take advantage of the partial label learning framework which let us train our classifiers on a *set* of candidate labels per transcriptomic profile. This approach overcomes the need for a training dataset annotated at single-cell resolution. We show that we obtain classification accuracy similar to the fully supervised case. We explore the effect of varying the amount of partially labeled data and of considering the hierarchical structure of the label set (derived from the developmental processes) in the models on simulated and real biological datasets.

## Introduction

The field of developmental biology aims to unravel the mechanisms that underlie the transformation of a single-cell into a complex multicellular organism. This process encompasses molecular, cellular, and tissue-level dynamics. To measure this diversity of processes, several complementary acquisition methods are available [1]. In particular scRNASeq methods make it possible to establish a transcriptomic profile for each cell, i.e. a measure of the quantity of RNA associated with each gene [2]. In practice, this transcriptomic profile is, for each cell, a vector of high dimension (generally of the order of 20,000: the number of genes considered), such that each of the coordinates of this vector represents the number of RNA molecules measured associated with a given gene. Because of their high dimensionality, scRNAseq datasets cannot be explored by hand. Therefore several methodological approaches have been proposed to make sense of these datasets. Particularly, several nonlinear dimension reduction methods have been developed, with the aim of preserving the structure of the data and allowing the identification of subcategories with a biological rationale [3–5]. In addition, a number of methods aim at reconstructing how these data points evolve over time, through the task of trajectory inference [6, 7]. Moreover, some works have sought to infer the structure of these datasets by making explicit their link with the spatial organization of cells [8]. Overall, scRNASeq datasets have proven very useful to answer a variety of questions in particular in developmental biology [9, 10]. The main remaining limitations come from the fact that transcriptomic profiles do not usually contain temporal and precise spatial information [11], and manual annotation of individual cell profiles can be tedious, time-consuming and require expert knowledge [12].

Because of the rapid accumulation of data, one of the major challenges currently is to perform accurate automated annotation of single-cell RNASeq datasets into accurate cell types [12–15]. Annotation strategies rely on the establishment of large atlases and expert knowledge, as well as transfer learning strategies for the use of already annotated datasets to newly generated ones. While often gene markers can be used to identify unambiguously a given cell type, in some cases, the annotation remains partial with an ambiguity that cannot be resolved by known *a priori* information [9].

We propose in this paper to take advantage of partial label learning strategies for the problem of automated single-cell transcriptomics profile annotation. The idea of partial label learning [16] consists in the task of multiclass classification, where, instead of a single label per sample, a *set* of candidate labels is given per sample, among which only one is correct. One of the application of this approach consists in the case when even with expert knowledge on known marker genes, it is not possible to perform single-cell annotation as it is the case for *C. elegans* [9]. The second class of applications concerns the situation where a reference atlas exists, providing sufficient information for establishing sets of candidate labels for each transcriptomic profile, but not sufficient for a single-cell resolution annotation (because of time, cost, and computational limitations). In this paper, we establish and address the challenge of resolving partially annotated single-cell transcriptomics profile by solving a partial label learning problem. We compare various implementations of the problem that we test both in simulated data and experimental data. We focus in particular on the question of labels with a hierarchy, such as the ones that can be found in developing and differentiating systems [12, 17–21]. This leads us to a benchmark of partial hierarchical labeling methods adapted from classical approaches to the problem of partial labels.

From the machine learning point of view, the problems we want to solve are hierarchical classification problems with partially labeled training data. Partial label learning has been studied in a variety of real-world learning scenarios, including automatic face naming [22, 23], web mining [24], and multimedia content analysis [23]. Researchers have proposed various approaches to tackle the partial label learning problem using well-known machine learning techniques. These techniques include maximum likelihood estimation [25], convex optimization [16], k-nearest neighbors [22, 25] and more. While maximum margin methods (e.g. SVM) are a powerful solution to such a partial labeling setting [26, 27], we will investigate prototype-based methods, nearest neighbor, logistic regression and ensemble methods as well to provide a comparative study of a wide spectrum of methods.

Similarly, hierarchical classification has been addressed by many approaches in the machine learning community and in particular for extreme classification (hierarchical classification with a very large number of labels) [28]. Hierarchical classification has been intensively studied for image data [29] and textual data [30]. Yet the research in this area has mainly focused on specific application contexts for these data, large-scale problems (in the number of classes and in the number of data) [28–32], hierarchical multilabel classification, discovery of a hierarchy underlying the data [30]. In comparison, few works have been concerned with learning from small datasets and/or with partial label in a framework similar to ours [33, 34]. For example, [33, 34] proposed an EM-like approach but dedicated to textual data and for a limited number of labels organized in a very flat hierarchy (of height equal to 2 or 3).

We will investigate and compare several methods from various families of approaches to solve our classification problems on biological data: a SVM-based approach (e.g. [35, 36]), a prototype-based approach originally inspired from [37], a kNN-based approach, logistic regression [38] and ensemble methods such as Random Forest [39] and Extreme Gradient Boosting Methods [40, 41]. We will consider three different settings: standard supervised multiclass classification, multiclass classification with partially labeled data, hierarchical classification with partially labeled data. While the methods are naturally available for the standard supervised multiclass classification setting, we will use variants to the two other settings, when available, or propose an extension otherwise.

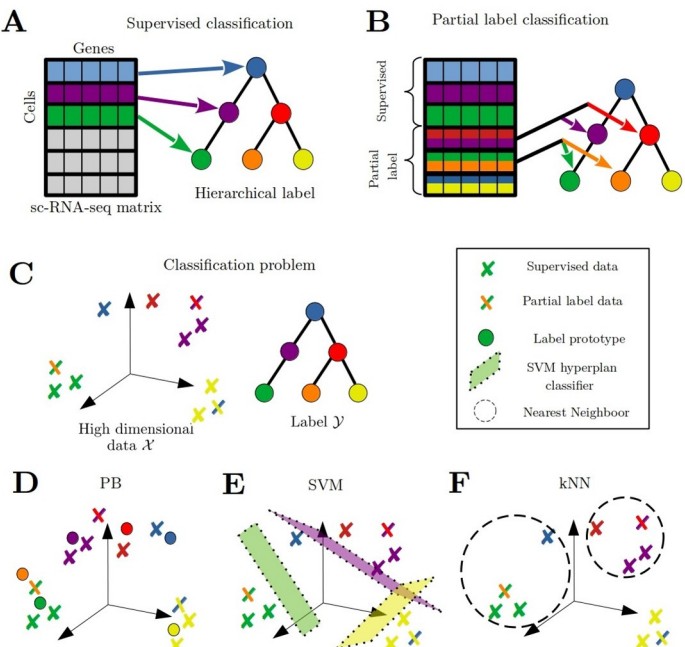

**Fig 1. Problem statement and classification methods.** A) Classification of transcriptomic profiles with a hierarchical label structure. Each training sample is associated to a unique label. B) Partial label learning framework, the transcriptomic profiles can be annotated with one or a *set* of candidate labels. C) In practice, the samples lie in a high-dimensional space, a proportion of the training samples is partially labeled, i.e. labeled with a set of candidate labels (illustrated with two colored crosses), while the remaining training samples have a unique label (and a unique color). The hierarchical structure of the labels is represented as a tree. D-F) Schematic illustration of some of the various methods explored in this study for the problem of partial label learning. We detail their extension to partial label learning and hierarchical learning in the Methods section. D) The Prototype-Based (PB) approach method consists in learning a projection of the data $X$ and label $Y$ into a new common embedding space. The embeddings of the labels are their associated prototypes, and the projection of the samples is learned such that the samples are as close as possible to their associated label's prototype. For prediction, the labels of test samples will be the one of the closest prototype in the common embedding space. E) The Support Vector Machine (SVM) approach consists in learning hyperplanes as classifiers in the initial feature space $\mathcal{X}$. The prediction for a new test sample will depend on the position of this sample with respect to the learned hyperplanes. F) The k-Nearest Neighbors (kNN) algorithm predicts a new test sample's label by first identifying the nearest training samples in $\mathcal{X}$ and then using a majority vote to identify the most common label among them.

## Materials and methods

### Task formalization

The problem of automatically classifying partially labeled transcriptomic profiles is summarized in Fig 1. The task we are interested in may be viewed as a multi-class classification problem where samples $x \in \mathcal{X}$ belong to a set of labels $\mathcal{Y}$ that may be organized in a hierarchy. Here we consider $\mathcal{X} = \mathbb{R}^d$ and we note $|\mathcal{Y}| = c$. We explored several settings, that we briefly describe now.

**Partially labeled setting.** The *partially labeled* setting corresponds to the case where the training set includes samples which are labeled as well as samples whose labeling is incomplete (or partial). We will distinguish between the two kinds of samples as fully labeled samples and partially labeled samples. The supervision for a partially labeled sample, say $x_i$, is a label set $Y_i \subset \mathcal{Y}$ which is known to include the true label of $x_i$, and whose cardinal is limited (e.g. 2 to 10 in our experiments).

Generally, one wants to learn to classify input samples based on a training set $D$ which consists of the union of two datasets; a fully supervised dataset $D_s$ and a partially labaled dataset $D_{pl}$:

$$D = \{(x_i, y_i) \in \mathcal{X} \times \mathcal{Y}, i = 1, .., l\}$$
$$\cup \{(x_i, Y_i) \in \mathcal{X} \times \mathcal{P}(\mathcal{Y}), i = l + 1, .., l + m\} \equiv D_s \cup D_{pl} \quad (1)$$

where $\mathcal{P}(\mathcal{Y})$ represents the set of subsets of $\mathcal{Y}$.

**Fully supervised hierarchical classification.** Another important characteristic of the datasets we are interested in lies in the hierarchical organization of the labels (e.g. lineage hierarchy, as in [9]). It has been shown in the machine learning literature that taking into account such prior knowledge may be beneficial in various hierarchical multiclass classification approaches [36, 37, 42]. Note that hierarchical classification often comes with a very large number of labels, this type of classification problems is named extreme classification [28]. As usually done in hierarchical classification, we will consider this prior information on labels through a matrix $C$, such that for two labels $y_1, y_2 \in \mathcal{Y}^2$, $C_{y_1, y_2}$ stands for a measure of dissimilarity between the two labels (e.g. the shortest path in the hierarchy, which is a tree, between two nodes/labels).

Many approaches have been proposed for multiclass hierarchical classification in the *fully supervised* setting [29, 36, 37], which are extensions of traditional multiclass methods, mainly Support Vector Machines [36] and Prototype-based methods [29, 37]. The formalization of the hierarchical classification problem is often cast as the minimization of the average dissimilarity between the predicted and true labels.

**Partially labeled hierarchical classification.** While it has been rarely explored to our knowledge in the machine learning community, the real datasets we are interested in include both aspects, partial labeling and a hierarchical organization of the labels.

## Learning strategies

We investigated several families of methods: *Support Vector Machines based methods (SVM)*; *Logistic Regression (LR)*, *Nearest Neighbor methods (kNN)*, *Prototype-based methods (PB)* and *ensemble-based methods* (Random forest (RF) and Extreme Gradient Boosting Methods (XGBM)). We first explain in a generic way, independently of the classification model, how we handled the partially supervised setting. We then detail how we handled the hierarchical information to perform hierarchical classification. We note $f_W$ a classification model (e.g. a SVM) parameterized by a set of parameters $W$, $f_W(x)$ the class output by model $f_W$ for a sample $x$, $score(f_W, x, c)$ the score for model $f_W$, class $c$ and sample $x$ (in particular $f_W(x) = \arg\max_c score(f_W, x, c)$). Finally, we note $loss(f_W, x, y)$ the loss computed for a model $f_W$ and a labeled training sample $(x, y)$.

To start with, in the basic supervised multiclass classification problem, when the dataset includes supervised samples only, the solution is usually handled by an optimization problem of the form:

$$\arg\min_W \quad \Omega(W) + \frac{\mu}{l} \sum_{i=1}^{l} loss(f_W, x_i, y_i) \quad (2)$$

where $W$ are the parameters of model $f$ to be learned, $\Omega(W)$ is a regularization term, and $\mu$ tunes the trade-off between the two terms of the objective function.

**Partially labeled setting.** The corresponding optimization problem consists of finding the optimal parameters $W$ and labeling $\tilde{y}_i \in Y_i$ of samples $(x_i, Y_i) \in D_{pl}$ with respect to the

following objective:

$$\arg\min_{W} \quad \left[ \Omega(f) + \frac{\mu}{l} \sum_{i=1}^{l} loss(f_W, x_i, y_i) + \frac{\lambda}{m} \sum_{i=1}^{m} \min_{\tilde{y}_{l+i} \in Y_i} loss(f_W, x_{l+i}, \tilde{y}_{l+i}) \right] \tag{3}$$

where $\mu$ and $\lambda$ tune the trade-off between the various terms of the objective.

**Iterative Refinement Learning (IRL).**   To deal with such an optimization problem, we used an EM-like iterative optimization algorithm (see Algorithm 1) which is inspired by proposals for learning SVM with latent variables (e.g. [27, 43]). We instantiate this algorithm here to fit our partially labeled setting. The algorithm consists, after initialization, of an iteration over two steps. The first one infers the labeling of partially labeled samples given the model's parameters, and the second one refines the model parameters using the inferred labels for partially labeled data, as in a fully supervised mode.

More precisely, in the first step, one infers a new guess $\tilde{y}_i^{(t)}$ for every sample $(x_i, Y_i)$ in $D_{pl}$, using the current model $f_{W^{(t-1)}}$, where exponent $(t)$ is used to name a variable at iteration number $t$. This is achieved by inferring the most likely label for each $x_i \in D_{pl}$ according to:

$$\tilde{y}_i^{(t)} = \arg\max_{y \in Y_i} score(f_{W^{(t-1)}}, x^i, y). \tag{4}$$

In the second step one builds a supervised version of $D_{pl}$, $D_{pl}^{(t)} = \{(x_i, \tilde{y}_i^{(t)}) \in \mathcal{X} \times \mathcal{Y}, i = l+1, .., l+m\}$. Finally, a new model $f_{W^{(t)}}$ is obtained by refining $f_{W^{(t-1)}}$ in fully supervised mode trained on $D_s \cup D_{pl}^{(t)}$.

The initialization may consist of random initialization of $\tilde{Y} = \{\tilde{y}_{l+i}, i = 1..m\}$ and a first learning of $f$ in a supervised mode on $D_s$ and on $D_{pl}$ using this labeling or it might be a supervised learning on $D_s$ only.

**Algorithm 1:** *Iterative Refinement Learning* (IRL) strategy for partially labeled setting.

```
Data: D = Ds ∪ Dpl
Initialize: Initialize model f⁽⁰⁾; t=1
while Optimization not complete do
   /* Infer labels for data in Dpl using current model (Eq (4)) */
   ∀i ∈ 〚1,m〛, ỹᵢ⁽ᵗ⁾ = arg maxᵧ∈Yᵢ score(f_W⁽ᵗ⁻¹⁾, xᵢ, y)
   /* Build a labeled view of Dpl */
   D_pl⁽ᵗ⁾ = {(xᵢ, ỹᵢ⁽ᵗ⁾) ∈ 𝒳 × 𝒴, i = l+1, .., l+m}
   /* Build a new fully supervised dataset for learning the model */
   D⁽ᵗ⁾ = Ds ∪ D_pl⁽ᵗ⁾
   /* Refine model f in a fully supervised mode with D⁽ᵗ⁾ */
   f_W⁽ᵗ⁾ = One step gradient reestimation of f_W⁽ᵗ⁻¹⁾ using supervised data-
   set D⁽ᵗ⁾
   t=t+1
end
```

**Iterative full retraining (IFR).**   Note that the IRL algorithm (Algorithm 1) is not well suited for all classification models. It can only be used for models whose optimization can be started from the solution learnt at the previous iteration. This is restricted to models whose optimization is performed iteratively with, for example, gradient-based optimization routines. In this case, the refinement step can be implemented with a single step of gradient descent at every iteration. However, this is not adapted for ensemble-based models, nor for nonlinear (kernel) SVM when optimization is performed in the dual space. For such cases, we investigated a variant that we detail in Algorithm 2 close to the one in [27]. It relies on a complete learning of the model at each iteration. Because of this, it is computationally expensive.

Moreover, we observed experimentally that the alternative training schema in Algorithm 2 is quite sensitive to the expressivity of the classifier used. If the classifier is too expressive it could be able to learn any labeling of $D_{pl}$ (for instance the first one $D_{pl}^{(1)}$) and because this labeling would never change, the algorithm would be inefficient. Specifically, we observed that training very expressive classifiers with this algorithm often yielded poor performances, this suggests restricting the classifier's expressivity to achieve reasonable results. More elaborated optimization routines might be used to reach better results if the model has to be trained until convergence at every iteration.

**Algorithm 2:** *Iterative Full Retraining* (IFR) learning strategy for the partially labeled setting.

```
Data: D = D_s ∪ D_pl
Initialize: Initialize model f^(0); t=1
while Optimization not complete do
  /* Infer labels for data in D_pl using current model (Eq (4)) */
  ∀i ∈ [[1, m]], ỹ_i^(t) = arg max_{y∈Y_i} score(f_w^(t-1), x_i, y)
  /* Build a labeled view of D_pl */
  D_pl^(t) = {(x_i, ỹ_i^(t)) ∈ X × Y, i = l + 1, .., l + m}
  /* Build a new fully supervised dataset for learning the model */
  D^(t) = D_s ∪ D_pl^(t)
  /* Learn model f from scratch in a fully supervised mode with D^(t) */
  f^(t) = Learn till convergence with the supervised dataset D^(t)
  t=t+1
end
```

**Inference in the partially labeled setting.** At inference time, partial labeling information may be available or not for a test sample, which leads to two inference strategies, with (Eq (5)) or without (Eq (6)) such prior information:

$$\hat{y}_i = \arg\max_{y \in Y_i} score(f_W, x_i, y) \tag{5}$$

$$\hat{y}_i = \arg\max_{y \in \mathcal{Y}} score(f_W, x_i, y) \tag{6}$$

**Hierarchical classification.** There has been several extensions of standard multiclass classification methods to hierarchical classification, the most popular are SVM-based and prototype-based methods whose optimization formulation involves inequality constraints such as the one used in SVM-based models:

$$\forall y \in \mathcal{Y} \setminus \{y_i\}, score(f_w, x_i, y_i) \geq score(f_w, x_i, y) + 1 \tag{7}$$

where the constraint means that for a given training sample, the score of the right class should be bigger than the score of any alternative class, plus a margin (equal to 1).

The hierarchical extension of such a method consists of leveraging the prior information on labels through a matrix $C$, such that for two labels $y_1, y_2 \in \mathcal{Y}^2$, $C_{y_1, y_2}$ stands for a measure of dissimilarity between the two labels. As has been done in several studies one may exploit this information to build hierarchical classifiers by using such a dissimilarity measure as a margin in constraints in Eq (7) yielding what is known as a *margin scaling* strategy [35]. The above constraints are changed to:

$$\forall y \in \mathcal{Y} \setminus \{y_i\}, score(f_w, x_i, y_i) \geq score(f_w, x_i, y) + C_{i,j} \tag{8}$$

Doing so, the score of a very distant class from the true one should be very low (because $C_{i,j}$ would be large) while the one of a very close class could be higher (but still below the score of the true label). It may be shown that by formulating the optimization problem with such inequalities in an SVM framework, one minimizes an upper bound of the average dissimilarity between the predicted and the true label [35].

Similarly, several works have studied the question of extending ensemble methods to hierarchical problems, however mainly in the context of multilabel classification [44, 45], which differs from our setting because a sample can belong to several classes. Due to the lack of off-the-shelf extension of ensemble methods for hierarchical classification, we will not include them in the experimental study for this particular context.

**Partially labeled hierarchical classification.**   This last setting is a mix of the two previous settings and may be dealt with using a combination of above strategies, for methods for which these strategies apply. We provide additional details in the S1 Text.

## Models

We experimented with several models and strategies. We distinguish between two families of models. On the one hand, we investigated the use of models that do not require full training at every iteration but which may be iteratively refined along iterations (with algorithm 1—*IRL*), this includes all models that are trained with an iterative algorithm (e.g. gradient descent) such as prototype-based methods (taking inspiration from [37]), linear SVM, logistic regression, and KNN as a special case since it does not require training at all. Note that we also investigated the use of nonlinear SVM with an approximation of the kernel [46, 47] that enables a gradient-based learning routine as well. On the other hand, we considered models whose training at one iteration should be complete (with algorithm 2—*IFR*) and does not benefit from the training at the previous iteration, such as Random Forests [39], Gradient boosting [41] and nonlinear (kernel) SVM [48]. The overall training strategy for these latter models is then much more costly than for the first set of models. All the methods and their implementation are summarized into Table 1.

More details on these methods and their implementation can be found in S1 Text. In particular, we detail the various SVM implementations that we used for investigating both linear and nonlinear SVM with both algorithms 1 and 2, and we provide more details on how we implemented prototype-based methods, SVM and kNN for the partial labeling problem and the hierarchical classification problem.

**Table 1. Details of methods implementation.**

| Methods | Acronym | Algo |
|---|---|---|
| Prototype-Based with | | |
| - linear projection | PB-l | *IRL* |
| - nonlinear projection (neural network) | PB-nn | *IRL* |
| Support Vector Machine | | |
| - linear | SVM | *IRL* |
| - nonlinear with kernel approximation | $\hat{k}$-SVM | *IRL* |
| Logistic Regression | LR | *IRL* |
| k-Nearest Neighbors | kNN | N.A |
| Random Forest | RF | *IFR* |
| Extreme Gradient Boosting Method | XGBM | *IFR* |

*IRL* stands for Iterative Refinement Learning, *IFR* stands for Iterative Full Retraining, *N.A* means not applicable.

## Datasets and experimental settings

We first describe the datasets on which we performed experiments to compare the various methods in different settings. We used simulated datasets using the prosstt library [49] and single-cell RNASeq reference datasets [9, 50, 51]. After describing the characteristics of the datasets, we describe the experimental settings.

**Simulated datasets.** We used the prosstt library to simulate single-cell RNA-seq data in various conditions [49]. We chose to work with this particular library because of its ability to simulate differentiation processes and implement differentiation trajectories with various topologies.

First, we design the topology of the label set. To highlight the impact of the topology of the label set on the classification results, we chose to generate three different scenarios:

- The *Linear* case corresponds to a succession of nodes where there is no division or bifurcation. The number of labels is in that case directly equal to the depth of the tree.

- The *Branches* case corresponds to an asymmetric situation where when there is a division or bifurcation, one branch will continue as the linear topology and the other one will continue dividing. The number of labels is in that case given by $\sum_{i=0}^{t+1} i$ where $t$ is depth of the tree.

- The *Binary* case corresponds to the extreme case where each branch is dividing into two new branches forming a binary tree. The number of labels is given by $\sum_{i=0}^{t+1} 2^i$ where $t$ is the depth of the tree.

These scenarios are illustrated in Fig 2.

In the prosstt library [49], the evolution of gene expression across the differentiation trajectories is modeled using genetic programs that are in practice random walks on a tree. One of the main hyperparameters is the number of intrinsic genetic programs, denoted $g$. Because having more genetic programs increases the combinatorics of gene expression in each branch, we anticipate that a higher number of programs would lead to more differences between the various parts of the tree, hence easier classification problems. We set this number to 10 and 50.

Finally, the last parameter we use to generate the datasets is the coefficient $\alpha$ which controls the characteristics of the negative binomial probability distribution that is used to generate realistic gene expression [49]. We chose $\alpha = 0.1$ which corresponds to low levels of noise and $\alpha = 0.5$ which corresponds to high levels of noise.

Using these three parameters (topology, $g$ and $\alpha$), we generated 12 datasets of approximately 300 000 samples per dataset, which we use in Figs 2, 3 and 4. To get comparable results, we fix the number of labels to 255 for the *Linear* and *Binary* topologies and to 253 for *Branches*.

**Experimental datasets.**

**C. elegans.** The first experimental dataset comes from an atlas of development for the nematode *C. elegans* at single-cell resolution [9]. It initially consists of 89701 vectors of dimension 20222, where each coordinate corresponds to the expression level of a gene. We reduced the dimension to 3842, by selecting a representative subset of genes corresponding to transcription factors obtained from the gene ontology extracted from Wormbase [52]. The labels have a tree structure that is derived from the fact that they encode the successive cell divisions in the embryonic development of the model organism *C. elegans* from the initial cell to the adult. If we consider the complete tree, there are 1342 possible labels [53] which are the cell names in the cell division history. Among the 89701 transcriptomic profiles, 5143 have been annotated with a unique label (6% of the whole dataset), 42631 have been partially annotated with at least two labels (47% of the whole dataset) and 41927 have no labels (47% of the whole

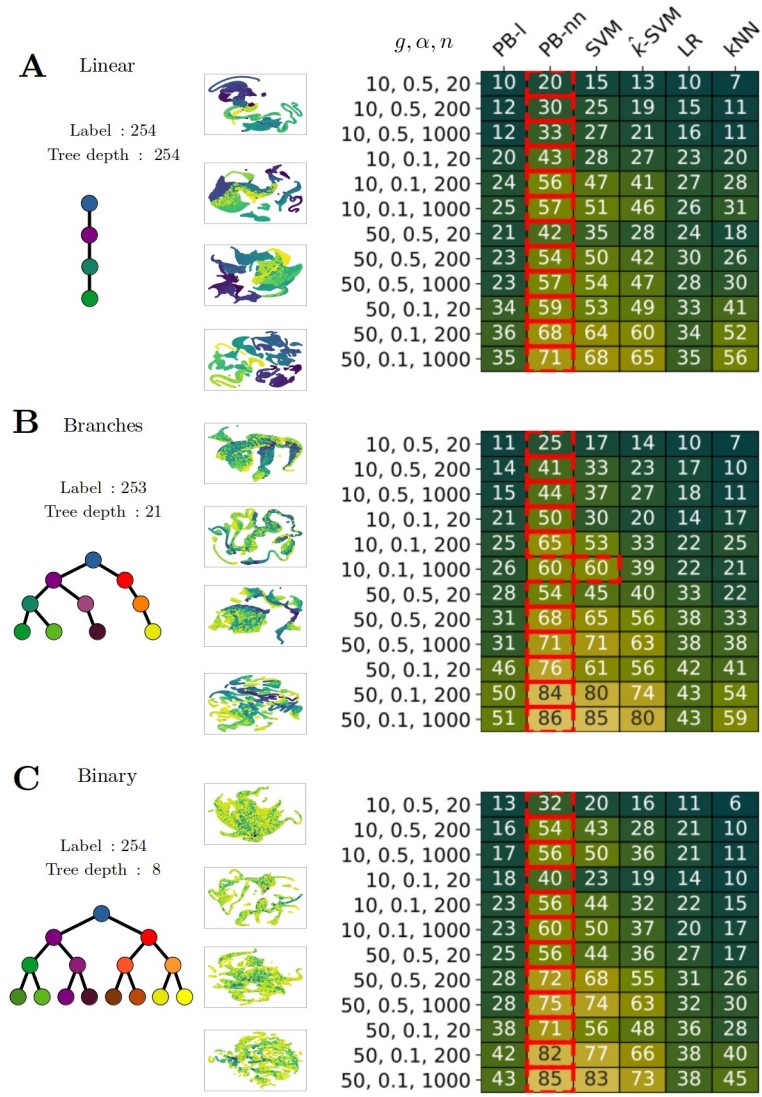

**Fig 2. Supervised performance on simulated data.** A-B-C) The first column represents the three main topologies studied with simulated scRNASeq data. The second column shows the UMAP projections of the various datasets [4], the color is plotted according to pseudotime, which is computed during the data simulation. Heatmap of performance according to the method and dataset and number of samples. Each row represents a dataset which is named by order, according to the number of programs (g), the coefficient ($\alpha$) used to control the noise in the simulations, and the number of samples (n) for each label in the training data. Each column shows a different method (as presented in the Material and Methods section). Each square corresponds to the percentage accuracy between the true label and the prediction using the given method. The red squares indicate the best-performing method per row, i.e. per experimental setting, using a t-test and as significance criteria p-value $< 0.1$.

dataset). In our experiments, we used only the 5143 transcriptomic profiles that have a unique label, the associated labels cover 95 labels among the 1342 labels of the whole hierarchy.

**S. mediterranea.**   The second experimental dataset comes from an atlas of development of the planaria *S. mediterranea* at single-cell resolution. We use as label hierarchy, the lineage tree obtained in [50] using the PAGA method [54]. The dataset consists of 21612 transcriptomic vectors of dimension 28065, distributed among 51 different labels. In order to have a uniform distribution of transcriptomic profiles per labels we subsampled the set of samples associated

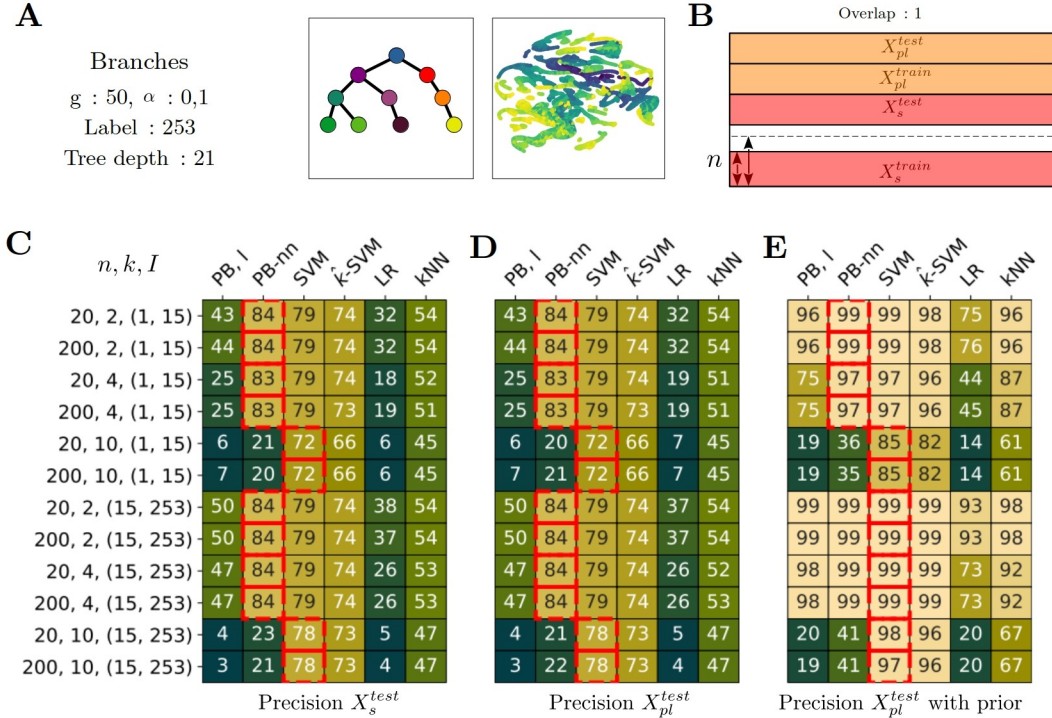

**Fig 3. Partial label results for the setting with label overlap (o = 1).** A) Dataset *Branches*, 50 genetic programs, low level of noise. UMAP of the transcriptomic profiles, colored according to the pseudotime. B) Partial label scenario. Full overlap between supervised label and partial ones. For supervised subset, we train on $n = 20$ or $n = 200$ examples per label and 200 examples for each partial label. C,D,E) Precision according to the setting. Each row corresponds to a setting depending on, by order, I: the interval for partial label sampling, k: the number of partial label set, n: the number of training examples per label supervised. The $i^{th}$ row of C corresponds to the $i^{th}$ row of D and E. C)Precision on $X_s^{test}$. D) Precision on $X_{pl}^{test}$. E) Precision on $X_{pl}^{test}$ when prior information is available for test too. C-E) The red squares indicate the best-performing method per row, i.e. per experimental setting, using a t-test and as significance criteria p-value < 0.1.

to labels have the highest number of examples (such as "neoblast1", which has 6343 examples initially). This leads us to 7595 transcriptomic profiles. In that case, the lineage tree is not deep, most branches have a length between 2 and 3 labels and the largest is 6. The original dataset is available at https://shiny.mdc-berlin.de/psca/.

***Myeloid progenitors.*** The third experimental datasets comes from an atlas of hematopoiesis at single-cell resolution [51]. This dataset is available at https://www.ncbi.nlm.nih.gov/geo/query/acc.cgi?acc=GSE72857GSE72857. It contains 2730 transcriptomic profiles of dimension 10719. Data are distributed among 26 labels in total.

Following standard practice, PCA projection was applied to the three datasets using the first 100, 50, and 100 components for *C. elegans*, *S. mediterranea* and *Myeloid progenitors*, respectively. For the two last datasets, PCA components were directly obtained from public available sources [21].

**Partially labeled datasets.** To investigate the performance of our approaches in various partial label settings we generated a collection of partially labeled datasets on which to test the various methods. From the fully supervised datasets described above, we generated a collection of partially labeled datasets $D = D_s \cup D_{pl}$ corresponding to different settings in terms of the proportion of labels for which we have fully supervised training samples, amount of partial labels, overlap between labels of supervised samples and of partially labeled samples, and balance between supervised samples and partially labeled samples.

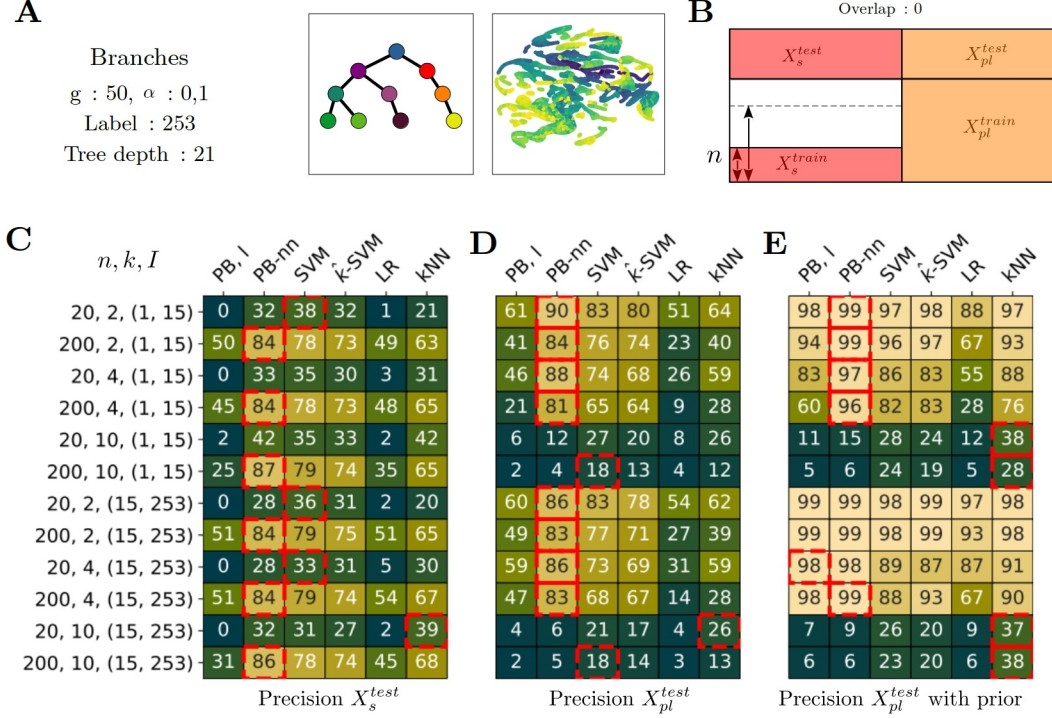

**Fig 4. Partial label results for the setting with no label overlap (o = 0).** A) Dataset *Branches*, 50 genetic programs, low level of noise. UMAP of the transcriptomic profiles, colored according to the pseudotime. B) Partial label scenario. No overlap between supervised label and partial ones. For supervised subset, we train on $n = 20$ or $n = 200$ examples per label and for the partial label subset, we select 200 examples for each partial label. C,D,E) Precision according to the setting. Each row corresponds to a setting depending on, I: interval for partial label sampling, k: number of partial label set, n: number of training examples per fully supervised labels. The $i^{th}$ row of C corresponds to the $i^{th}$ row of D and E. C) Precision on $X_s^{test}$. D) Precision on $X_{pl}^{test}$. E) Precision on $X_{pl}^{test}$ when prior information is available for test too. C-E) The red squares indicate the best-performing method per row, i.e. per experimental setting, using a t-test and as significance criteria p-value < 0.1.

## Experiments

To characterize the various experimental settings, we first introduce a set of characteristics for the datasets below. In the following, we note $\mathcal{Y}_s$ the set of labels that occur in $D_s$ and $\mathcal{Y}_{pl}$ the set of labels that occur in $D_{pl}$.

1. **Percentage of shared labels between $D_s$ and $D_{pl}$.** We generated datasets with different distributions of labels in $D_s$, in $D_{pl}$ and in their intersection.

   - *o* is the percentage of unique labels that belong to the intersection $\mathcal{Y}_s \cap \mathcal{Y}_{pl}$ with respect to the number of unique labels in $\mathcal{Y}_s \cup \mathcal{Y}_{pl}$. It is defined as $o = \frac{|\mathcal{Y}_s \cap \mathcal{Y}_{pl}|}{|\mathcal{Y}_s \cup \mathcal{Y}_{pl}|}$.

2. **Number of supervised samples**. We define two parameters to control the amount of supervision when training the models.

   - *n* is the number of fully supervised samples per label $\alpha \in \mathcal{Y}_s$. This parameter is used only for simulated data.

   - *p* is the percentage of fully supervised samples used for training data for each label $\alpha \in \mathcal{Y}_s$. This parameter is used for the experimental datasets because of the non uniformity of sample distribution over the set of labels.

3. **Characteristics of the partial labels**. Choosing the way we build partial label $Y_i$ subsets has a clear impact on the model training. We used two parameters:

- $k$ is the number of possible labels for a given partially labeled sample, i.e. for a training sample $x_i \in D_{pl}$, it is the cardinal of the subset $Y_i$, $|Y_i|$. We keep it constant for all data in $D_{pl}$.

- $I$ is an interval of minimum and maximum distances, to the true label $y_i$ of a sample $x_i$, within which labels will be randomly chosen to belong to a subset $Y_i$.

**Scenarios.** We considered a number of experiments in specifically designed experimental settings which are characterized by the set of parameters $s = (o, p, n, k, I)$ discussed above. These numerical experiments are organized around two main scenarios which are illustrated in Figs 3 and 4, panel B.

1. Complete overlap between the labels of supervised and of partially labeled samples (i.e. $o = 1$) (Fig 3B). In that scenario we investigated various experimental settings by varying $n$ or $p$, $k$ and $I$.

2. No overlap between the labels of supervised and partially labeled samples (i.e. $o = 0$) (Fig 4B). In that scenario, we investigated various experimental settings by varying $n$ or $p$, $k$ and $I$.

**Tasks.** For each of the scenarios and the experimental settings we considered two tasks that allow investigating in deep the behaviors of our methods.

1. *Fully supervised setting*. The model is trained on fully supervised data only.

2. *Partially labeled setting*. The model is trained on both fully supervised and partially labeled data.

**Metrics.** We report experimental results according to two metrics of interest that we detail here. We note $n_{tot}$ the number of samples which are used for computing the criterion. The first metric is computed based on label prediction without prior information using Eq (6) while the second metric is computed based on label prediction with prior information using Eq (5).

- *Precision*: the precision is the percentage of correct classification, it is defined on a set of test data whose true label is known as $\frac{1}{n_{tot}} \sum_i 1_{\hat{y}_i, y_i}$.

- *Precision with prior*: this criterion is defined for partially labeled samples only $(x_i, Y_i)$. It is defined as the precision when the prediction is performed within the label set $Y_i$.

**Generating partial label data and Train Test split.** Data generation follows a multi-step process:

1. Initially, the data is split based on the label and whether there is label overlap. If $o = 0$, the data is divided evenly between two sets. In contrast, if $o = 1$, all labels are included in both sets. A test set is created for each label group, with a proportion of 20% of the samples allocated for testing.

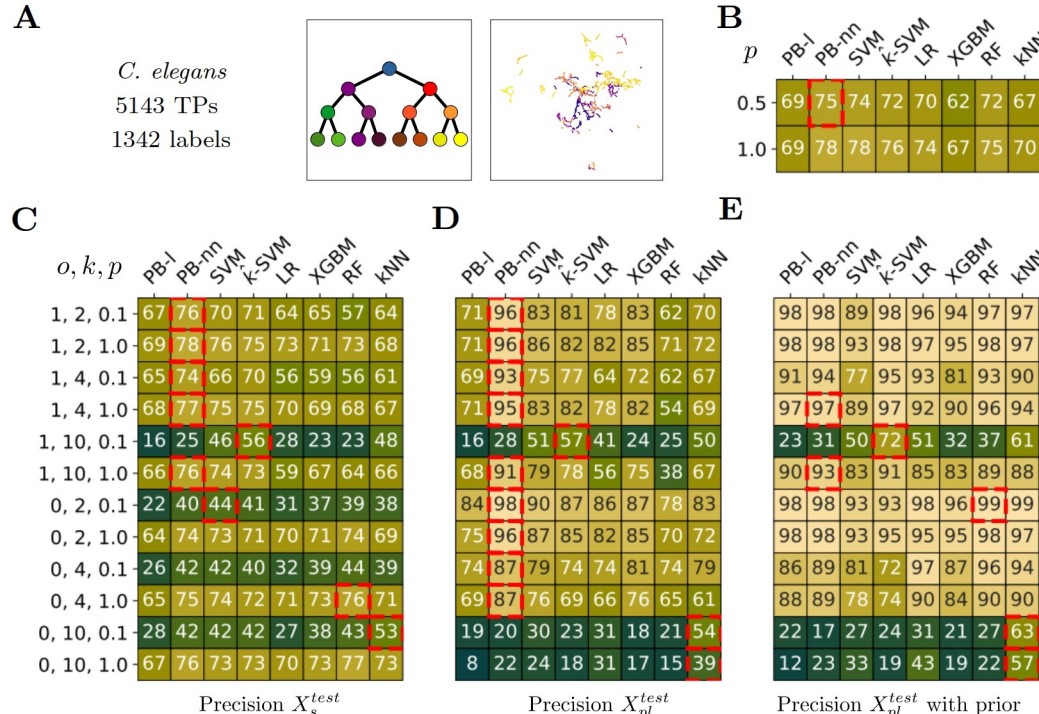

**Fig 5. Classification performance on the *C. elegans* dataset.** A) *C. elegans* dataset main features. The topology of the label corresponds to the *Binary* one. We use the entire lineage tree [53] for label classification. The UMAP projection of the annotated transcriptomic profiles, colored according to their respective label. B) Performance of our methods in a fully supervised setting according to the proportion $p$ of supervised training data. 0.5 means that models are trained with half of the training data. C,D,E) Precision according to the setting in partial label scenarios. Each row corresponds to a setting depending on, by order, $o$: Overlap between supervised and partial label, $k$: number of partial label set, $p$: proportion of fully supervised training data. The $i^{th}$ row of C) corresponds to the $i^{th}$ row of D) and E). The median value of pairwise label distance is 16 and we chose $I = (1, 8)$ for partial label sampling. C) Precision on $X_s^{test}$. D) Precision on $X_{pl}^{test}$. E) Precision on $X_{pl}^{test}$ when prior information is available for test too. B-E) The red squares indicate the best-performing method per row, i.e. per experimental setting, using a t-test and as significance criteria p-value $< 0.1$.

2. The supervised training set is further subdivided based on the proportion ($p$) value, which can take on values of 0.1 or 1.0, or the number ($n$) of training examples, set to either 20, 200, or 1000. Stratified train-test splits are used for this process. For the partial label training set, a fixed number of 200 examples is chosen in the case of the simulated datasets (Prosstt *Branches*), and 80% percent of the data points in the training set is selected in the case of the experimental datasets.

3. The partial labeling is generated as follows:

   - For each label, we create a set of pairwise probable labels to increase the co-occurrence, with the number of pairs set arbitrarily to 15 within the specified interval $I$. In the simulated dataset presented in Figs 3 and 4, we selected a close distance interval (1, 15) and a far interval (15, 253). However, for the experimental datasets, we opted for only close intervals, directly determined by the number of labels. These details and information can be found in Figs 5, 6 and 7.

   - For each partially labeled sample, the partial labeling is sampled uniformly from the associated label co-occurrence set. It contains $k = 2$, 4, or 10. This process is also performed for

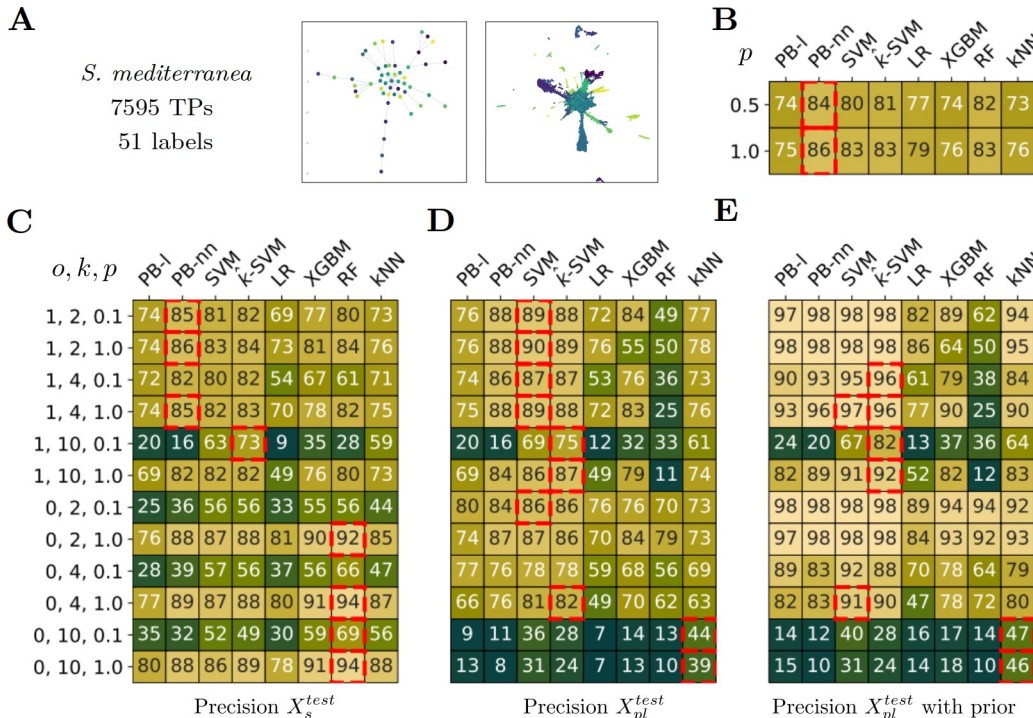

**Fig 6. Classification performance on *S. mediterranea* dataset.** A) *S. mediterranea* dataset main features. The topology of the label is close to *Branches* case. The UMAP of the annotated transcriptomic profiles, colored according to their respective labels. B) Performance of the methods in a fully supervised setting according to the proportion ($p$) of supervised training data. 0.5 means that the models are trained with half of the training data. C,D,E) Precision according to the setting in partial label scenarios. Each row corresponds to a setting depending on, by order, $o$: overlap between supervised and partial label, $k$: number of partial label set, $p$: proportion of fully supervised training data. The $i^{th}$ row of C) corresponds to the $i^{th}$ row of D) and E). The median value of pairwise label distance is 3 and we chose $I = (1, 4)$ for partial label sampling. C) Precision on $X_s^{test}$. D) Precision on $X_{pl}^{test}$. E) Precision on $X_{pl}^{test}$ when prior information is available for test too. B-E) The red squares indicate the best-performing method per row, i.e. per experimental setting, using a t-test and as significance criteria p-value $< 0.1$.

the partial label example test set, allowing the evaluation of prediction with prior information metric.

**Multiple runs, gridsearch and averaged results.** We report averaged results in the following. We explain how we get these now.

For every experimental setting, as defined by a set of values $set = (o, p, n, k, I)$, we first generate one dataset with the desired characteristics. Then we perform 5 cuts of the dataset into train and test $X_s^{train}$ and $X_{pl}^{train}$ on one side and $X_s^{test}$ and $X_{pl}^{test}$ on the other side (where $s$ stands for supervised, and $pl$ for partially labeled).

For every train-test cut, we use the train data to learn models and select hyperparameters since the learning of each model involves hyper-parameters. As is usually done in machine learning the optimal hyperparameter values are selected using cross validation on the training set (we used 5 fold cross validation), then the selected hyperparameters are used to learn a model on the full training set, which is then evaluated on the test set.

At the end, for each experimental setting, we get 5 performance values, one for each train-test cut, that are averaged to get one averaged performance value that we report on the figures.

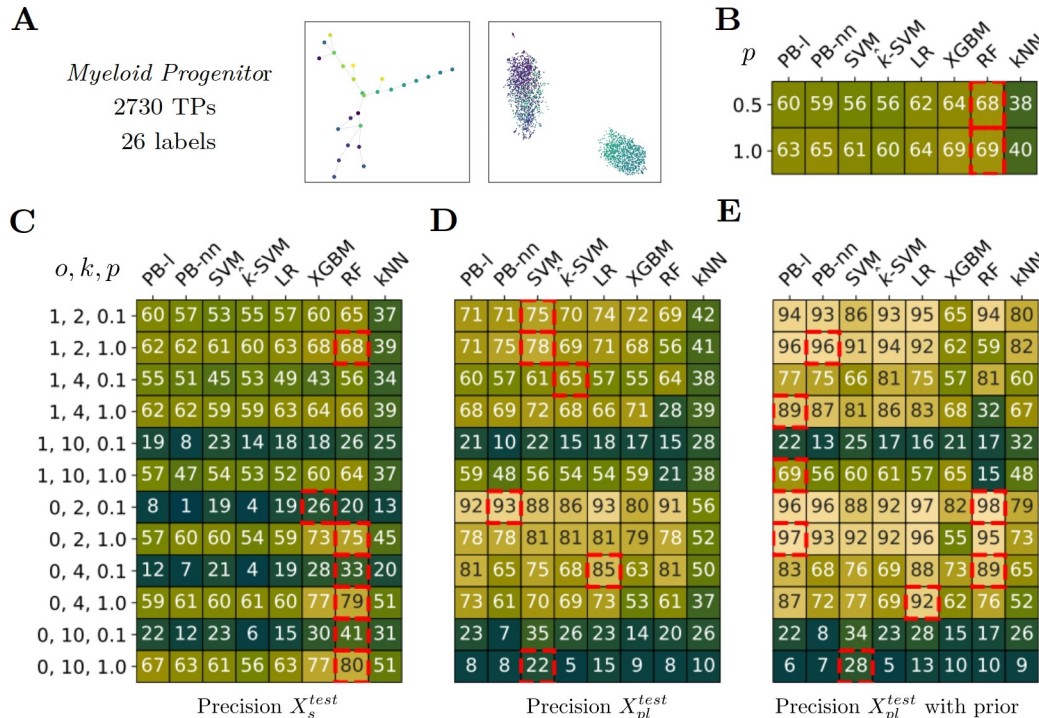

**Fig 7. Classification performance on *Myeloid progenitors* dataset.** A) *Myeloid progenitors* dataset main features. The topology of the label is close to the *Branches* case. The UMAP projection of the annotated transcriptomic profiles, colored according to their respective labels. B) Performance of the methods in a fully supervised setting according to the proportion ($p$) of supervised training data. 0.5 means that models are trained with half of the training data. C,D,E) Precision according to the setting in partial label scenarios. Each row corresponds to a setting depending on, by order, $o$: overlap between supervised and partial label, $k$: number of partial label set, $p$: proportion of fully supervised training data. The $i^{th}$ row of C corresponds to the $i^{th}$ row of D and E. The median value of pairwise label distance is 4 and we chose $I = (1, 4)$ for partial label sampling. C) Precision on $X_s^{test}$. D) Precision on $X_{pl}^{test}$. E) Precision on $X_{pl}^{test}$ when prior information is available for test too. B-E) The red squares indicate the best-performing method per row, i.e. per experimental setting, using a t-test and as significance criteria p-value < 0.1.

**Statistical significance.** For each experimental setting, given a row in a figure, we evaluated which one among all the methods had the best performance. We perform t-test between the first and second best methods. When several methods are available in a same family (as SVM, PB or kNN), we select the best version in the family (linear vs nonlinear in Figs 2, 3, 4, 5, 6 and 7, or hierarchical vs flat in Fig 8). For example, if, for a given experimental setting, PB-nn was the best between PB methods and linear SVM was the best between SVM methods, we performed a paired t-test between those two. If one of the two methods performed significantly better than the other, with a p-value < 0.1, we reported it on the figure by drawing red square with a dotted line around the best performance value. We chose to set the significance threshold at 0.1 for the p-value because the data are noisy and there are only 5 performance values per experimental setting and per method.

## Results

First, we analyze the fully supervised setting on 12 artificial Prosstt datasets, as shown in Fig 2. We then focus on one specific simulated dataset to explore different settings of the partial label learning problem. The different settings include scenarios with label overlap (Fig 3), and

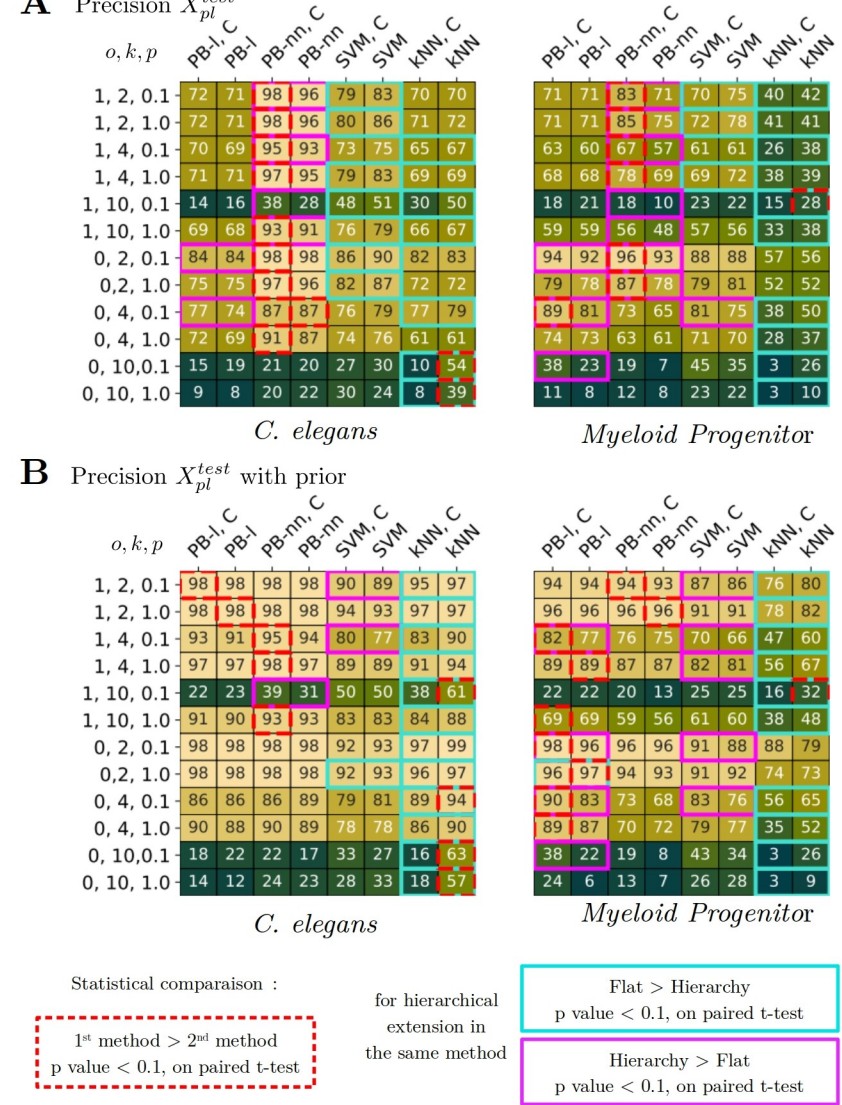

**Fig 8. Including hierarchical information improves performance.** A-B) Comparison of each method with its hierarchical version, where we take into account $C$ the pairwise distance matrix between the labels in the margin computation. We compute paired t-test and highlight results when hierarchical model are performing significantly better than their non-hierarchical counterparts in magenta, when it's the opposite, in blue. We also report in red, which one of the methods performs significantly better for each experimental setting. A) Precision on $X_{pl}$ on *C. elegans* (left table) and *Myeloid progenitor* datasets (right table). B) Precision on $X_{pl}$ with prior information on *C. elegans* (left table) and *Myeloid progenitor* datasets (right table).

scenarios without label overlap (Fig 4). Furthermore, we analyze the performance of our approaches on three real-world datasets, *C. elegans* [9], *S. mediterranea* [50] and *Myeloid progenitors* [51] in Figs 5, 6 and 7 respectively. Finally, we compare the performance of hierarchical and flat methods in Fig 8 where we focus on partial labeling cases with different settings on two real-world datasets. For the artificial datasets which contain about 300,000 samples each, we only compare scalable methods, the ones exploiting the *IRL* algorithm. When considering the real datasets, which are much smaller, we provide a comprehensive comparison of all methods.

## Supervised setting

Fig 2 displays the results for all methods on 12 different datasets.

## The dataset topology influences classification accuracy

For each topology, we order datasets by decreasing difficulty: few genetic programs (g=10) with noise ($\alpha$=0.5), without noise ($\alpha$=0.1), then more genetic programs (g=50) with noise ($\alpha$=0.5) and without noise ($\alpha$=0.1). In Fig 2, for the dataset corresponding to 50 programs, 0.1 noise and 20 examples per label in the training set, the method Prototype-Based neural network (PB-nn) achieves 0.59 accuracy for *Linear* topology, 0.76 for *Branches* and 0.71 for *Binary* For the Support Vector Machine (SVM), the results are respectively 0.53, 0.61 and 0.56.

## PB-nn and SVM methods are well adapted for classification

The general behavior on all datasets is that Prototype-Based with Neural Network (PB-nn) performs significantly better (p value < 0.1) than linear SVM, which is better than linear Prototype-Based (PB-l), Logistic Regression (LR) and kNN, which are similar. In details, we take kNN as a baseline, we first observe that PB-l performs slightly better in most of datasets especially with few training examples (20 samples per label), in particular for *Binary*, 10 programs, 0.1 noise, 20 training examples per label; the accuracy is 0.18 (PB-l) versus 0.10 (kNN). However, with 50 genetic programs, less noise and more examples, PB-l tends to be less efficient. The contrast becomes more prominent when comparing PB-nn and SVM to kNN, with accuracy potentially doubling or even tripling depending on the settings. As an example, if we take a difficult dataset like *Binary*, 10 programs, little amount of noise ($\alpha$=0.1) and a low number of training examples per label (n = 20), the accuracies are 0.40 (PB-nn), 0.23 (SVM) and 0.10 (kNN). Finally, when the dataset becomes easier, the accuracy becomes very high: 0.86 (PB-nn), 0.85 (SVM) and 0.59 (kNN), when looking at dataset *Branches*, 50 programs, a small amount of noise ($\alpha$=0.1) and a lot of training examples per label (n = 1000). We explain the contrast between PB-nn and PB-l by the intrinsic complexity of sc-RNA seq data, which requires nonlinear projection to achieve good results. We also observed that kernel-SVM performs worse than the regular SVM and hypothesize that this decreased accuracy comes from the kernel approximation. More discussion can be found in S2 Text.

## High accuracy with few training samples

We are interested in assessing the influence of the number of training samples per label. Because of the lack and limitation of supervised data in real-world settings, we want good performance even if only a small training dataset is available. The general value varies between 10 and 200 examples per label for several real-world scRNASeq datasets. We observe that SVM works well with 200 examples per label, PB-nn works well with 20 examples and the performances with 200 examples are close to SVM with 1000 examples. We compare *Branches*, 10 programs, 0.5 noise: with 200 training examples per label, PB-nn accuracy is 0.41, whereas the accuracy for SVM is 0.33 and with 1000 training examples, SVM's accuracy is 0.37. Similarly for *Linear*, 10 programs, 0.5 noise, 200 training examples per label, PB-nn achieves 0.3 and SVM 0.25, while the accuracy of SVM increases to 0.27 with 1000 training examples.

To summarize, our supervised approaches are relevant for simulated transcriptomic data and we find that both SVM and PB-nn perform very well on all datasets. Moreover, the PB-nn

significantly outperforms SVM, LR and kNN in 35 experimental settings out of 36, making it the method of choice for this task. Finally, we highlight that PB-nn has high predictive power even with a small amount of training data.

### Partial label, Overlap = 1

In this paragraph, we study the partial label problem with label overlap. It means that for a given label, the training set contains both fully supervised samples and partially labeled ones. We focus the study on one dataset: *Branches*, 50 programs, a small amount of noise ($\alpha$=0.1) as shown in Fig 3.

### Performances are close to fully supervised setting

On both $X_s$ and $X_{pl}$, the methods perform very well. Particularly, we found that with 20 supervised plus 200 partially labeled examples, with a set of candidate labels of size k = 2 or 4, the methods PB-nn and SVM have accuracy of 0.84 and 0.79 versus 0.84 and 0.80 in the fully supervised cases (with 200 fully supervised training examples per label). The performance on the partial label test set is identical to the supervised test set, contrary to the case of no label overlap as shown below. Note that for both PB-l and LR, performances were below other methods on this dataset on the fully supervised case (in Fig 2, accuracies were 50 and 43 respectively with $n$ = 200 samples).

### With prior information, the results are excellent

When partial label information is also available for the test set, (precision with prior), PB-nn and SVM have 0.99 accuracy when k = 2, 0.97 with k = 4. It means that, the method can be used to disambiguate new data with a small amount of information.

### SVM works very well with difficult partially labeled settings

When the learning task becomes extreme, for example, 20 supervised examples and 200 partial label ones with a large number of candidate labels (k = 10), the SVM still performs well: 0.72 and with prior information accuracy goes to 0.85. Whereas the accuracy of PB-nn falls with a value of 0.2 and 0.36 with prior information.

### The distance between labels influences the partial learning performance

When the candidate labels are far from the real label in the partial labeling information ($I$ = (15, 253)), it leads to better results. Especially, with prior information, looking at PB-nn, for k = 4, and 20 supervised examples, the accuracy goes from 0.97 to 0.99. The parameter $I$ plays a role in partial labeling, as it can impact the results. For example, when selecting two labels situated at the same depth in a lineage tree, they will have identical pseudotime values (distance to the root), but their hierarchical distance might be large.

To summarize, the methods perform very well with partially labeled data. The performances are comparable to the fully supervised setting. We show that just a few amount of supervised data is needed to reach top performance and we provide some criteria to improve the performance according to the characteristics of the candidate label set (cardinal and distances).

### Partial label, Overlap = 0

This time, we study the scenario when there is no overlap between supervised label and partial label. This task is much more difficult than the previous one. We still focus on the same data-set: *Branches*, 50 programs, a small amount of noise ($\alpha$=0.1) as shown in Fig 4.

### The precision on supervised data depends on the size of the training dataset

We observe that the accuracy on the supervised test set is sensitive to the quantity of fully supervised examples. Indeed, when there is only n = 20 supervised examples per label in the training set, the accuracy is at 0.32 for PB-nn (k = 2, I=(1,15)), but it increases to 0.84 with n = 200 training examples. This is due to the fact that the model will focus more on the partially labeled training data when there is less supervised data. We can also observe that the corresponding accuracy for the partial label test set decreases from 0.9 to 0.84 for the same configuration. Depending on the task, the methods require a compromise between supervised and partial label training data. For example, it could be relevant to increase the number of supervised examples for supervised prediction whereas this could decrease the quality of prediction on partially labeled data.

### Ability to predict partial labels only

This time, we are characterizing the ability to predict the good label for partially labeled data when there is no supervised data at all. It means that the model can learn to predict good labels without having seen it in the supervised training data. When the partial label set is not too large (up to k = 4), the best performances are obtained with PB-nn: (for n = 20, k = 2 and I= (1,15)) the accuracy reaches 0.9 and (for k = 4), it goes to 0.88 while the corresponding SVM performances are 0.83 and 0.74. When the setting becomes extreme, (k = 10), the performances fall but are still better than random. In this situation, SVM stays reliable.

### Results are excellent with prior information

As before, models can predict the good label even without supervised training example. When the size of partial labels is not too large, PB-nn reaches 0.99 (k = 2) and 0.97 (k = 4), while SVM obtains 0.97 (k = 2) and 0.82 (k = 4). As in the previous paragraph, when the distance between labels increases ($I$ = (15, 253) in the partial labeling accuracy goes from 0.96 to 0.99 for PB-nn (k = 4 and n = 200)).

Overall, we demonstrate that the models trained on partially labeled data only can predict the right label with high accuracy even without supervised training data on this label. With prior information, the models have performances close to perfect accuracy. We suggest using PB-nn methods while the size of the set of candidate labels is small (k = 4) performances are significantly better (p-value < 0.1). For more difficult situations, SVM is significantly better (p value < 0.1) and achieves very good results. Overall we find, for simulated data in the partially labeled case, that, out of 72 experimental settings, PB-nn performs significantly better than all the other methods in 40 cases (i.e. 55% of the time), while SVM outperforms all the other methods in 26 cases (i.e. 36%) of the cases.

### Real-world datasets

In the following paragraphs, we apply an identical pipeline to each of the three experimental datasets. Because of the relatively smaller size of the experimental datasets compared to the artificial ones, we now include, in addition to the methods studied in the previous section, alternative methods such as ensemble-based approaches that can only be optimized with the

costly Algo 2. All related results are presented in Figs 5, 6 and 7. Our analysis begins by examining the classification problem in a supervised setting (panel B in the figures). Afterwards, we explore the two scenarios of partial labeling, one with label overlap and the other without label overlap (panel C in the figures). First, we provide an analysis of the common results observed across the datasets, we then delve into the specific characteristics and nuances of each dataset.

## Common results

In the supervised setting, we observed that PB-nn performs the best for the *S. mediterranea* and *C. elegans* dataset. For *Myeloid progenitors* dataset, Random Forest (RF) achieves the best results. We note that all methods perform well even when they have only half of the training examples ($p = 0.5$). In total, we found that PB-nn significantly outperforms all the other methods in 3 cases out of the 6 settings and RF 2 cases out of the 6.

For the partial label learning setting, specifically when there is label overlap ($o = 1$), the precision on the supervised test set is nearly as high as the performance achieved in the fully supervised setting. Moreover, in the partial label test set, we noticed even higher performances, even when using $k = 10$. However, it is important to have a sufficient amount of supervised training data ($p = 1$) to achieve such results.

On the other hand, in scenarios without label overlap ($o = 0$), we observed a behavior similar to the one observed on simulated data, with a compromise between supervised and partial label performances. As the amount of supervised training data increases ($p = 1.0$), the performances on the supervised set increase, while the performances on the partial label set decrease. In certain cases, results are excellent, particularly when $k = 2$ and $k = 4$. Sometimes, the performance in this setting even surpasses that of the fully supervised setting, which can be attributed to the smaller size of the test set.

When examining the performance with prior information available, we observe that the Prototype-Based (PB) methods stand out for $k = 2$, across all the datasets. As the value of $k$ increases to 4 or 10, the SVM methods remain reliable and maintain competitive performance.

Overall, in the partial label setting, we find that the PB family of methods significantly outperforms all the other methods 22 times out of 108 experimental settings (20% of the time). On the other hand, the SVM family of methods significantly outperforms the other methods 22 times among the 108 experimental settings (20% of the time), RF outperforms in 14 cases, (13% of the time), kNN methods are significantly better in 11 cases out the 108 experimental settings (10% of the time). Finally, LR and XGBM outperform both in one case.

Both of the ensemble methods (RF and XGBM) rely on Algo 2. We observe that in general these methods tend to be more efficient in simpler scenarios. Specifically, the performances in fully supervised learning tasks (panel B of the figures) are competitive in all three real-world datasets. In partial label learning tasks, these methods tend to focus more on supervised data (especially with no overlap ($o = 0$)) and achieve good results on the corresponding test set, but performances decrease on the partial label data. The main limitations may come from the algorithm itself; particularly, if the model is too expressive, it might learn the first random inference of labels, and if the model is too simple, performance will be limited. More discussion on the Algo 2 can be found in S2 Text.

Generally, in both partial label learning scenarios (with and without overlap), we consistently observed excellent results, particularly when the proportion of supervised data is low (p = 0.1). These findings once again underscore the value and justification for utilizing partially labeled data for classification. The remarkable performance achieved with a limited amount of supervised information further validates the effectiveness and practicality of employing partial label learning methods in this context.

### *C. elegans*

This dataset follows a *Binary* topology with a depth of approximately 10/11, comprising about 1342 labels. The dataset we are working with covers 95 labels scattered throughout the lineage tree.

In the fully supervised setting, both the PB-nn and SVM methods exhibit the best performance, achieving an accuracy of 0.78 when all the training data are available ($p = 1.0$).

When dealing with partial label data and label overlap ($o = 1$), we observe excellent results for both the supervised and partial label test sets. Even with a higher level of uncertainty ($k = 10$), all methods prove effective as long as there is sufficient supervised data available ($p = 1.0$).

In the more challenging scenario without label overlap ($o = 0$), the performances remain excellent until $k = 4$ and remain high for $k = 10$, even for the PB method, which has shown to be slightly less efficient in this situation.

Additionally, we find that kernel SVM is particularly relevant for inference with prior information, especially in cases of label overlap ($o = 1$), where it outperforms the regular SVM method. The kernel approximation seems to enhance the predictive capabilities of SVM, especially in situations with more complex hierarchical relationships.

### *S. mediterranea*

This dataset exhibits a relatively flat hierarchy, and its topology resembles that of the *Branches* dataset, comprising several concise linear differentiation processes. The intrinsic complexity of the dataset appears to be effectively captured by both the PB-nn and SVM methods.

In the supervised setting, we find that the performances of all the methods are quite close (PB-nn stays significantly better). Both PB-nn and SVM obtain great results across most settings. Notably, in the label overlap scenario ($o = 1$) with $k = 10$ and ample supervised training data ($p = 1.0$), PB-nn achieves an accuracy of 0.84.

Additionally, we also observe that kernel-SVM is efficient in handling two of the most challenging settings, ($k = 10$, $o = 1.0$), with either a limited or a complete amount of supervised data ($p = 0.1$ and $p = 1.0$). It shows superior performance on both the supervised and partial label test sets.

In the case of no overlap ($o = 0$), the common behavior, observed across all the methods is to focus on supervised data. It is even more remarkable for RF. Notably, when all the supervised training data are available ($p = 1.0$), performances of RF outperform all the other methods, for $k = 2, 4, 10$ with an accuracy of 92, 94, 94 respectively. The performances on the partial label data are however quite below.

### *Myeloid progenitors*

The hierarchy is much more present in this dataset. It consists of three main linear differentiation processes. There is a small number of labels (26) and a low amount of training data. We notice that PB-nn performs less than PB-l due to overfitting. Indeed, in the supervised setting, PB-nn demonstrates better performance when all the training data is available ($p = 1$) compared to using only half of it ($p = 0.5$).

In partial label learning task, PB methods and SVM outperform kNN significantly except when there is not enough data ($p = 0.1$) or if it's too difficult ($k = 10$). For the performance on the partial label, PB-nn stands out until $k = 4$, especially with label overlap ($o = 1$). However, the lack of data combined with the complexity ($k = 10$) seems to lead to overfitting which is the limit of the method. A solution could then be to reduce the number of hidden layers in the neural network.

In the case of no label overlap ($o = 0$), we find the same results as in *S. Mediterranea* in Fig 6. When all the supervised training data are available ($p = 1.0$), RF outperforms in the supervised test data for both $k = 2, 4, 10$ with an accuracy of 75, 79, 80 respectively. It indicates one more time, that when there is relatively less information in the partially labeled data, the model focuses on the supervised data.

## Incorporating hierarchy improves performance

Including the hierarchical structure of the label set significantly enhances the performance of models. To present the results more concisely, we first described in the previous section the results for flat methods which do not take into account hierarchical structures between labels. Now, we aim to compare these flat methods to a hierarchical approach. For the artificial datasets, we found that the hierarchical prototype-based performs equally well as flat-PB, and for both SVM and kNN, flat models were slightly better than hierarchical ones. We will now focus only on the experimental datasets. In the supervised setting, we find that hierarchical models perform with similar performance as Prototype-Based non-hierarchical models, slightly underperform for SVM-type models, and similarly for kNN models. However, in the partial label setting, hierarchical methods stand out and greatly improve performance. Fig 8 illustrates the methods alongside their hierarchical extensions, and we conducted paired t-tests to highlight instances where hierarchical models perform significantly better than their non-hierarchical counterparts (p-value < 0.1). Moreover, we found that the PB-nn method with hierarchy significantly outperformed all other methods (SVM and kNN in this experiment) in 25 cases among all the 48 experimental settings (52% of the time) and more generally the PB family of methods (flat, hierarchical, linear and nonlinear) significantly outperforms all other methods in 30 cases which represents 62% of the cases.

## Prediction for partial label test set

The first panel of Fig 8, demonstrates that hierarchical PB methods consistently outperform their non-hierarchical counterparts. For instance, in the case of *C. elegans* (with $o = 1$, $k = 2$, $p = 0.1$), the performance increases from 0.96 to 0.98. This improvement is even more significant in challenging scenarios (such as when $o = 1$, $k = 10$, $p = 0.1$), where the boost for *C. elegans* is from 0.28 to 0.38, and for *Myeloid progenitors* it is from 0.10 to 0.18.

For the SVM, the increase is not systematic, however, it can be detected in some of the cases. In particular, for *Myeloid progenitors*, in very difficult settings ($o = 0$, $k = 10$, $p = 0.1$), the accuracy increases from 0.35 to 0.45.

## Boost for prediction with prior information

The hierarchical PB-nn method performs better than PB-nn (which was initially the best method for *C. elegans*). Similarly, hierarchical PB-l outperforms PB-l in *Myeloid progenitors*. The hierarchical method PB-l highlights the following improvements for no overlap setting ($o = 0$, $p = 0.1$): ($k = 2$) from 0.96 to 0.98, ($k = 4$) from 0.83 to 0.90, and ($k = 10$) from 0.22 to 0.38.

Additionally, hierarchical SVM performs better when a smaller proportion of supervised data is available ($p = 0.1$). In the case of *C. elegans*, the hierarchical method highlights the following improvements for label overlap situation ($o = 1$): ($k = 2$) from 0.89 to 0.90, ($k = 4$) from 0.77 to 0.80, and ($k = 10$) remains at 0.5. For *Myeloid progenitors*, the hierarchical boost for no overlap situation ($o = 0$) is as follows: ($k = 2$) from 0.88 to 0.91, ($k = 4$) from 0.76 to 0.83, and ($k = 10$) from 0.34 to 0.43. In the case of label overlap situation ($o = 1$): ($k = 2$) from 0.86 to 0.87, ($k = 4$) from 0.66 to 0.7, and ($k = 10$) remains at 0.25.

Finally, we find that for both SVM and PB, the hierarchical extension can significantly improve performance compared to the flat version, especially when a small amount of supervised training data is available.

## Discussion

We proposed several approaches for the classification of partially labeled scRNASeq data. We provided detailed description of the optimization algorithms used to learn the models. To thoroughly evaluate the performance of all the methods, we devised a comprehensive testing protocol that covered various partial labeling scenarios. This protocol was applied to both simulated and experimental datasets. The experiments demonstrated that our approach, specifically the Prototype-Based Neural Network (PB-nn), achieved remarkably high accuracy in a fully supervised setting. Equally impressive was the models performances when only partial label training samples were available, showcasing the models' ability to leverage partial information effectively. Furthermore, we showed that incorporating partially labeled training samples corresponding to certain labels improved the performance on labels for which supervised samples were also available at training time. Our findings also provided insights into the effect of the level of uncertainty in partial labeling, allowing us to achieve the best performance. Ultimately, we found that incorporating hierarchical information could significantly increase performance in partial label settings, particularly in extreme situations with low number of training samples. Our results, showing that we can perform multiclass classification with high accuracy on partially labeled data can be associated to the fact that transcriptomic datasets are highly structured and lie on a low dimensional manifold [3]. Indeed when training a classifier on several transcriptomic profiles, even if they're partially annotated, will necessary take advantage of their geometrical properties. Because we showed that the highest performing partially labeled classifier is a nonlinear Prototype-Based method, we understand that the relationship between cell type identity and position in the feature space is not a linear function. Altogether we found that including the right amount of complexity in the model will lead to a sufficient amount of expressivity to learn the nonlinear distributions of transcriptomic profiles. Moreover, we showed that including the hierarchical structure of the label set as *a priori* information could help improve the models' accuracy. This is in line with the idea of learning the underlying geometry of the set of transcriptomic profiles and suggests moreover that the distribution of transcriptomic profiles in space is consistent with the label set hierarchical structure. Therefore when the number of samples is low, it will be more difficult to learn the underlying geometry of the data as there is less data points to sample the underlying manifold. Using the hierarchical structure of the label set will compensate by helping reduce the size of the parameter space for the models.

In this work, we focused on one aspect of the structure of the data, namely the hierarchical structure of the label set. We chose to focus on this aspect because of the hierarchical nature of differentiation during development [12, 18]. We envision that our approach could benefit from lineage information obtained directly at the level of data acquisition as it is the case with lineage tracing methods [19]. Moreover, other underlying aspects of the dataset could be influencing the structure of the data. In particular, recent developments in spatial transcriptomics indicate that the spatial organization of cells within a tissue plays an important role in transcriptomic state [8, 55]. Extensions of our approaches could take advantage of this spatial information when it exists. Similarly, recent experimental assays have led to combining multiple -omics [56], automated classification methods could also take advantage of these heterogeneous sources of data to infer cellular identity.

Altogether, we propose an approach that helps reduce the time spent on precise annotation of training datasets by considering sets of candidate labels per sample. In an era where transcriptomic data is being generated increasingly [57], being able to automatize more and with less expert knowledge the task of automatic cell type classification is highly relevant, as the quality of the annotation is currently the bottleneck [15]. Moreover, in addition to the automatic cell type classification task, one of the current challenges is to be able to combine multiple studies and atlases. Our results suggest that we could for example be using the classification results obtained from other already existing annotated datasets as a way to construct partial annotation for a new datasets. Similarly, we could be using tissue level information which is at scale high than cellular by nature and use all of the cell types expected in this tissue as a candidate label set. In conclusion, the flexibility and adaptability of the partial label learning framework is a key step in making transcriptomic single-cell data annotation easier and the corresponding prediction even more routinely used to interrogate the nature of the mechanisms underlying cell differentiation.

## Supporting information

**S1 Text. Supplementary methods.**
(PDF)

**S2 Text. Supplementary results.** Comparison between *IRL* and *IFR* algorithms and additional results on kernel SVM.
(PDF)

## Acknowledgments

We used OpenAI. (2021). ChatGPT: Large-scale generative models for natural language conversations. https://platform.openai.com/models/chatgpt and *Deepl Write* for text editing.

We thank Vincent Bertrand, Antoine Barrière, Pierre Recouvreux and Khulganaa Buyannemekh for fruitful discussions on the biology of *C. elegans*.

## Author Contributions

**Conceptualization:** Malek Senoussi, Thierry Artieres, Paul Villoutreix.

**Data curation:** Malek Senoussi.

**Formal analysis:** Malek Senoussi, Thierry Artieres, Paul Villoutreix.

**Funding acquisition:** Paul Villoutreix.

**Investigation:** Malek Senoussi.

**Methodology:** Malek Senoussi, Thierry Artieres, Paul Villoutreix.

**Resources:** Malek Senoussi.

**Software:** Malek Senoussi.

**Supervision:** Thierry Artieres, Paul Villoutreix.

**Validation:** Malek Senoussi.

**Visualization:** Malek Senoussi.

**Writing – original draft:** Malek Senoussi, Thierry Artieres, Paul Villoutreix.

**Writing – review & editing:** Malek Senoussi, Thierry Artieres, Paul Villoutreix.

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
