## [Decision Letter · Decision Letter 0]

7 Dec 2023

Dear Dr Villoutreix,

Thank you very much for submitting your manuscript "Partial label learning for automated classification of single cell transcriptomic profiles" for consideration at PLOS Computational Biology. As with all papers reviewed by the journal, your manuscript was reviewed by members of the editorial board and by several independent reviewers. The reviewers appreciated the attention to an important topic. Based on the reviews, we are likely to accept this manuscript for publication, providing that you modify the manuscript according to the review recommendations.

The reviewers appreciate the approach presented in the paper. However, they suggest to compare it to other methods to demonstrate its superior performance. These comparisons should be included in a revised manuscript.

Sincerely,

Stacey D. Finley, Ph.D.

Section Editor

PLOS Computational Biology

Reviewer's Responses to Questions

**Comments to the Authors:**

Reviewer #1: Senoussi et al. have performed the partial label single cell annotation to overcome the herculean task of specifying the labels for each cell. They have used different machine learning (Prototype based, SVM, and KNN) methods for partial automated single cell labeling on simulated and real world datasets. They identified that non-linear prototype based works best both in simulated and real world dataset and had a comparable accuracy with the fully supervised dataset. The manuscript has been well written and analyzed well. I have some comments:

The authors have been focused on the hierarchy multi class classification which is good for developmental biology. Does the partial also perform better when there is no hierarchy in the single cell dataset?

I appreciate that the authors have detailed their method section. But they should keep the method concise in the manuscript but can keep the elaborated methods in the supplementary.

In the figures, the brown square is not properly visible for the best performing method. Please use a different color.

PB, SVM, and Knn are good methods. I would suggest authors to use Tree based methods also like Random Forest, XGBoost, and LGBM. I think it will improve the accuracy. Also please try multiclass logistic regression.

It is good that the author has used both regular SVM and kernel SVM. I am a bit surprised that K-SVM has less F1 score than the regular SVM in single cell annotation. it could be due to the hyperparameter for K-SVM is not properly optimized, since in single cell data is non-linear. The same partial annotation works well for non-linear prototype based. Please explain.

Reviewer #2: I would like to see a comparison between the performance of your program and some others because I find it very useful to see if the program works better than others.

**Have the authors made all data and (if applicable) computational code underlying the findings in their manuscript fully available?**

Reviewer #1: Yes

Reviewer #2: Yes

PLOS authors have the option to publish the peer review history of their article (what does this mean?). If published, this will include your full peer review and any attached files.

Reviewer #1: No

Reviewer #2: No

Figure Files:

Data Requirements:

Reproducibility:

References:

---

## [Decision Letter · Decision Letter 1]

18 Mar 2024

Dear Dr Villoutreix,

We are pleased to inform you that your manuscript 'Partial label learning for automated classification of single-cell transcriptomic profiles' has been provisionally accepted for publication in PLOS Computational Biology.

Best regards,

Stacey D. Finley, Ph.D.

Section Editor

PLOS Computational Biology

Stacey Finley

Section Editor

PLOS Computational Biology

Reviewer's Responses to Questions

**Comments to the Authors:**

Reviewer #2: Congrats!!!

**Have the authors made all data and (if applicable) computational code underlying the findings in their manuscript fully available?**

Reviewer #2: Yes

PLOS authors have the option to publish the peer review history of their article (what does this mean?). If published, this will include your full peer review and any attached files.

Reviewer #2: No

---

## [Editor Report · Acceptance letter]

1 Apr 2024

PCOMPBIOL-D-23-01207R1 

Partial label learning for automated classification of single-cell transcriptomic profiles

Dear Dr Villoutreix,

I am pleased to inform you that your manuscript has been formally accepted for publication in PLOS Computational Biology. Your manuscript is now with our production department and you will be notified of the publication date in due course.

With kind regards,

Olena Szabo
